# Hemostatic Powders in Non-Variceal Upper Gastrointestinal Bleeding: The Open Questions

**DOI:** 10.3390/medicina59010143

**Published:** 2023-01-11

**Authors:** Omero Alessandro Paoluzi, Edoardo Troncone, Elena De Cristofaro, Mezia Sibilia, Giovanni Monteleone, Giovanna Del Vecchio Blanco

**Affiliations:** 1Gastroenterology Unit, Department of Systems Medicine, University Tor Vergata, 00133 Rome, Italy; 2Gastroenterology Unit, Department of Medical Sciences, University Tor Vergata Hospital, 00133 Rome, Italy

**Keywords:** ankaferd blood stopper, CEGP-003, endoclot, gastrointestinal bleeding, hemospray, hemostatic powders, hemostatic procedures, TC-325, UI-EWD, upper gastrointestinal bleeding

## Abstract

Hemostatic powder (HP) is a relatively recent addition to the arsenal of hemostatic endoscopic procedures (HEPs) for gastrointestinal bleeding (GIB) due to benign and malignant lesions. Five types of HP are currently available: TC-325 (Hemospray™), EndoClot™, Ankaferd Blood Stopper^®^, and, more recently, UI-EWD (Nexpowder^TM^) and CEGP-003 (CGBio™). HP acts as a mechanical barrier and/or promotes platelet activation and coagulation cascade. HP may be used in combination with or as rescue therapy in case of failure of conventional HEPs (CHEPs) and also as monotherapy in large, poorly accessible lesions with multiple bleeding sources. Although the literature on HP is abundant, randomized controlled trials are scant, and some questions remain open. While HP is highly effective in inducing immediate hemostasis in GIB, the rates of rebleeding reported in different studies are very variable, and conditions affecting the stability of hemostasis have not yet been fully elucidated. It is not established whether HP as monotherapy is appropriate in severe GIB, such as spurting peptic ulcers, or should be used only as rescue or adjunctive therapy. Finally, as it can be sprayed on large areas, HP could become the gold standard in malignancy-related GIB, which is often nonresponsive or not amenable to treatment with CHEPs as a result of multiple bleeding points and friable surfaces. This is a narrative review that provides an overview of currently available data and the open questions regarding the use of HP in the management of non-variceal upper GIB due to benign and malignant diseases.

## 1. Introduction

Upper gastrointestinal bleeding (UGIB) is mostly of non-variceal origin and has an annual incidence rate from 50 to 150 per 100,000 adults [1,2,3]. Peptic ulcer disease (PUD) is the most common cause of non-variceal UGIB (NVUGIB) and accounts for at least 50% of UGIB. Other frequent benign conditions underlying NVUGIB include gastroduodenal erosions (8–15%), Mallory–Weiss tears (8–15%), and erosive esophagitis (5–15%) [1]. Malignancy causes comprise 1–5% of all NVUGIB [4,5,6]. GIB is the initial presenting symptom of malignancy in up to 70% of patients [4,5], and the disease is already metastatic in about one-third of them [5]. Gastric cancer is the most common cause of malignancy-related NVUGIB (MR-NVUGIB) [4,5,6,7], and about one-third is metastatic [4]; the duodenum is the most frequent site of malignancy involving the small bowel [4,5], usually primary or metastatic from a pancreatic or biliary malignancy. In all NVUGIB related to benign and malignant lesions, upper digestive endoscopy (EGDS) with hemostatic endoscopic procedure (HEP) is the first-choice approach. Conventional HEPs (CHEPs) include injection agents (epinephrine, ethanol, and cyanoacrylate), contact thermal devices (heater probes and multipolar electrocautery probes), noncontact thermal devices (argon plasma coagulation), and mechanical devices (hemostatic graspers, band ligators, clips, and loops) [3]. The use of one or more CHEP, often in combination, induces endoscopic hemostasis in 67–100% of cases, although rebleeding occurs in up to about 25–30% of treated patients [2,3]. In recent years, hemostatic powder (HP) has been proposed and tested for the treatment of acute GIB, arousing growing interest among clinicians. The present narrative review provides an overview of currently available data and the open questions regarding the use of HP in the management of NVUGIB due to benign and malignant diseases.

## 2. Type, Modality of Action, and Application of Hemostatic Powder

To date, five HPs have been developed for the management of GIB (Table 1). The most diffused agents are TC-325 (Hemospray™; Cook Medical, Winston-Salem, NC, USA), EndoClot™ (EndoClot Plus, Santa Clara, CA, USA), and Ankaferd Blood Stopper^®^ (ABS; Ankaferd Health Products, Istanbul, Turkey); two more recent HPs are UI-EWD (NexPowder™; Nextbiomedical, Incheon, Republic of Korea) and CEGP-003 (CGBio, Seong-Nam, Republic of Korea). All these products are applied using a delivery catheter passed through the working channel of the endoscope to reach the bleeding site (Figure 1, Figure 2 and Figure 3).

Figure 1, Figure 2 and Figure 3: different lesions with active oozing bleeding and after hemostasis achieved by application of hemostatic powder.

TC-325 is compounded by bentonite, a naturally sourced aluminum phyllosilicate clay. This inert powder is propelled through a carbon dioxide pressurized catheter and sprayed at a distance of 1–2 cm from the bleeding site, forming a coat covering the lesion. This coat acts as a mechanical barrier and absorbs water, leading to a concentration of platelets and clotting factors with the activation of platelets and the coagulation cascade [8]. Due to its modality of action, TC-325 should be used in cases of active bleeding, as it is likely poorly or not effective in nonbleeding lesions [9]. Once sprayed over the area of bleeding, TC-325 adheres over the lesion for a limited time: the HP sloughs off the mucosa and is eliminated from the gastrointestinal tract within 24–72 h after application [8].

Endoclot is a starch-derived compound of hemostatic polysaccharides which, in contact with blood, absorbs water, causes a high concentration of platelets, red blood cells, and coagulation proteins at the bleeding site, and accelerates the physiological clotting cascade [8]. An air compressor provides consistent air pressure to propel the HP to the bleeding site. Endoclot also remains over the lesion for a limited time ranging from hours to days [10].

ABS is a plant-based agent that rapidly forms an encapsulated protein network providing focal points for erythrocytes and activated leukocyte aggregation. This network stems from interactions between ABS and blood proteins, such as fibrinogen, inducing protein agglutination. ABS also inhibits fibrinolysis and anticoagulant pathways, promoting wound healing [8].

UI-EWD is a biocompatible natural polymer produced using aldehyde dextran and succinic acid modified ε-poly (l-lysine). In the presence of water, the two compounds of the polymer react together, forming a hydrogel with multiple crosslinks resulting in high adhesiveness to the tissue. The hydrogel acts as a mechanical barrier and promotes hemostasis. UI-EWD is delivered via a system based on a liquid coating technology using a fluidized bed granulator. Since UI-EWD does not require clot formation to induce hemostasis, active bleeding is not necessary for this HP to be effective [11].

CEGP-003 powder is a mix of hydroxyethylcellulose and epidermal growth factor (EGF). Due to its adhesive and hygroscopic properties, hydroxyethylcellulose, in contact with water, forms an adhesive gel that acts as a barrier, while the EGF, by binding to EGF receptors, activates the syntheses of hyaluronan and aquaporin-3, both promoting wound healing [12,13].

### 2.1. Pros

HP is an attractive agent for endoscopic hemostasis in patients with GIB as it is straightforward to use, does not require prolonged training, can be applied to sites poorly accessible by endoscopy and hemostatic devices, can treat extensive areas with multiple bleeding points, and does not need to be in direct contact with the bleeding lesion. 

HP is generally safe and well tolerated. Some cases of embolization, bowel obstruction, and perforation have been reported in patients treated with Hemospray^®^, but, based on an analysis of the literature, the Food and Drug Administration (FDA) declared that Hemospray^®^ is an endoscopic hemostat with a very low-risk profile, with no contraindication except for gastrointestinal endoscopy (active or risk of gastrointestinal perforation) and gastrointestinal fistulas [14]. Endoclot also received a very low-risk profile statement from the FDA [15].

### 2.2. Cons


Clogging of the delivery catheter has been reported [16] during the release of TC-325, as it coagulates when in contact with fresh blood. During an emergency endoscopy with active GIB, it is necessary to aspirate blood from the lumen of the digestive tract, and the presence of blood in the working channel may determine the coagulation of HP, causing occlusion of the catheter. This issue may be overcome by a prolonged insufflation following blood aspiration to dry the working channel immediately before the spraying of HP [17]. Clogging seems to be infrequent (3.6%) with UI-EWD due to the system of delivery adopted [18]. Following the application of HP, the visibility of the target lesion is no longer guaranteed as the HP may obscure the endoscopic view.TC-325 has a high cost (US list price of USD 2500 in November 2020); this is the reason guidelines suggest that, in countries such as the United States, TC-325 should not be the initial modality used if other therapies can be readily applied [19].


## 3. Fields of Application of Hemostatic Powder in Nvugib and Evidence on Short- and Long-Term Efficacy

HP is used to induce hemostasis following failure of rescue therapy or together with CHEP (combination therapy) and also alone as a first-choice treatment (monotherapy) in upper and lower GIB related to malignant and benign conditions, including postendoscopic therapeutic procedures such as endoscopic submucosal dissection, sphincterotomy, and polypectomy, as well as to less frequent conditions such as Dieulafoy lesions and lesions due to graft-versus-host disease [17]. Although it has been used also in upper variceal bleeding and lower gastrointestinal bleeding, this article will only address the use of HP in NVUGIB. 

Table 2 lists studies on HP published to date, most of which investigate TC-325, likely because this agent is marketed more widely than the others; larger series have been published in recent years. For example, in the French “GRAPHE” registry including 202 patients with UGIB—most commonly related to PUD (37.1%), malignancy (30.2%), and postendoscopic therapy (17.3%)—TC-325 was used as rescue (53.5%) or first-line therapy (46.5%) and achieved an immediate hemostasis rate of 96.5%, with rebleeding in 26.7% and 33.5% of cases at day 8 and day 30, respectively, and the definitive hemostasis rate being 63% [20]. A retrospective nationwide study conducted in Spain involving 219 patients with UGIB—most frequently due to peptic ulcer (28%), malignancy (18.4%), and postendoscopic therapeutic procedures (17.6%)—showed that TC-325 was effective in inducing an immediate hemostasis rate of 93%, with rebleeding rates of 16.1%, 19.9%, and 22.9% at 3, 7, and 30 days, respectively, and a definitive hemostasis rate of 77% [21]. A multicenter European registry of 314 patients with UGIB—mainly due to PUD (53%), malignancy (16%), and postendoscopic procedures (16%)—treated with TC-325 reported immediate hemostasis in 89.5% of cases, with a rebleeding rate of 10.3% and a definitive hemostasis rate of 79.2% [22]. A retrospective study including 86 patients treated with HP as rescue therapy and monotherapy reported a high rate (88.4%) of immediate hemostasis but a cumulative rebleeding rate of 33.7% [23]. Overall, current evidence shows that TC-325 and other HPs can achieve immediate hemostasis in about 80–100% of UGIB regardless of etiology. In contrast, the rates of definitive hemostasis are more widely variable in the different studies, ranging from 40% to 100%. There may be several reasons for this discrepancy. Many published studies are retrospective analyses or case series, a small number are prospective investigations often lacking controls, and there is a paucity of randomized controlled trials (RCTs). Outcomes are variable: immediate hemostasis is almost uniformly defined as the stop of bleeding immediately after application of HP, whilst hemostasis is considered definitive (no relapse following application of HP) at different follow-up times (from 1 to 180 days); some studies evaluated the impact of HP on mortality rather than hemostasis. Lastly, the conditions underlying GIB in the study populations are highly heterogeneous and include GIB related to different malignancies, PUD with active bleeding (Forrest Ia and Ib) and with stigmata of recent but no-longer-active bleeding (Forrest IIa and IIb), postendoscopic therapy (e.g., endoscopic submucosal dissection and sphincterotomy), postsurgery bleeding, or other rarer lesions.

To better determine the efficacy of HP in GIB, two systematic reviews with meta-analysis were recently published. Facciorusso et al. [58] reviewed and included in a meta-analysis 24 studies, 3 of which were RCTs and 21 of which were retrospective investigations; 19 studies used TC-325, 4 used Endoclot, and 1 used CGEP-003, with a total of 1063 patients. Immediate hemostasis was achieved in 95.3% of patients, with a success rate of 91.9% in spurting bleeding; the rebleeding rate was 17.9% and 16.9% at 7 and 30 days, respectively, and, according to treatment strategy, the overall rebleeding rate was 13.5% and 24.8% in monotherapy and combined/rescue therapy, respectively. Although useful, this first meta-analysis has several limitations, such as the high heterogeneity of populations in the different studies (sample size and geographic areas), source of bleeding (PUD, other benign conditions, malignancies, and, in three studies, liver cirrhosis), study protocol (retrospective/prospective and time of rebleeding), and type of HP applied. A subsequent meta-analysis by Mutneja et al. [59] of 11 prospective studies (3 of which were RCTs) investigating different etiologies of GIB (2 on PUD only, 3 on variceal bleeding only, 2 on malignancies only, 3 on mixed benign and malignant tumors without variceal bleeding, and 1 on mixed conditions including variceal bleeding) in a total of 609 patients treated with TC-325 reported a pooled immediate hemostasis rate of 93.0% regardless of etiology and 95.3% in malignancy-related GIB (MR-GIB); the overall 12 h–30 days rebleeding rate was 14.4%, and the MR rebleeding rate was 21.9%. A separate analysis of the three RCTs on NVUGIB revealed that the probability of achieving immediate hemostasis was more than three times higher with TC-325 than with CHEP. However, statistical significance was not reached, likely due to the limited number of RCTs. Although the strength of this meta-analysis lies in the selection of only prospective studies using the same type of HP, the findings are to some extent underpowered by statistical and clinical heterogeneity due to the different sources of GIB, different rebleeding times, selection biases, and lack of controls.

A very recent noninferiority RCT compared the efficacy of TC-325 monotherapy versus CHEP (thermocoagulation or clipping with/without epinephrine injection) in 224 adult patients with NVUGIB due to malignancy and nonmalignant lesions [46]. The primary endpoint was control of bleeding within 30 days, defined as endoscopic hemostasis by the assigned treatment modality during the first endoscopy and no recurrent bleeding after endoscopic hemostasis. Secondary endpoints included failure to control bleeding during the first endoscopy and recurrent bleeding after hemostasis, combined with a lack of recurrent bleeding, with secondary outcomes including further interventions, transfusion, and death. Bleeding was controlled within 30 days in 90.1% of cases in the TC-325 group and in 81.4% in the CHEP group. Recurrent bleeding within 30 days did not differ between groups (8.1% versus 8.8%, respectively). Additional interventions, length of stay, and death were similar between groups. The limitations of this study were the heterogeneity of lesions, an unbalanced allocation of patients with malignancy, attending endoscopists not blinded to study treatments, and a low rate of Forrest Ia lesions.

Despite the limitations described above, data currently available indicate that HP can induce immediate hemostasis in the majority of treated patients, but rebleeding in the following days should be kept in mind. It has not yet been fully established whether the limited time of persistence of HP over the lesion may reduce long-term hemostasis and contribute to the occurrence of a late (7–30 days) rebleeding. Taking into account the time of elimination from the gastrointestinal tract, recent guidelines suggest the use of TC-325 as a temporizing intervention that should be followed by a second definitive HEP [60] in patients with persistent bleeding refractory to CHEPs [61] or conditionally in peptic ulcer bleeding [19].

## 4. Open Questions

### 4.1. Peptic-Ulcer-Related NVUGIB: HP Only as Rescue Therapy and Combination Therapy or Also as Monotherapy?

The efficacy of HP in combination or rescue therapy is well demonstrated, and this noncontact hemostatic technique should now be considered an indispensable part of the standard therapeutic armamentarium in emergency endoscopy. However, the use of HP as monotherapy means excluding CHEPs, which are routinely used in the endoscopy room and well known to be effective in stopping NVUGIB in 85–95% of cases, reducing the need for surgery and lowering mortality rates [47]. Therefore, before using HP as monotherapy instead of CHEPs, it should be certain that HP is highly effective in bleeding PUD. An early prospective single-arm pilot study by Sung et al. investigating the use of HP in NVUGIB-related PUD in 20 patients with Forrest Ia (1 patient) or Forrest Ib (19 patients) PUD revealed that TC-325 as monotherapy achieved immediate and definitive hemostasis rates of 95% and 89.5%, respectively [24]. In another prospective study of 20 patients with PUD, randomized to receive either TC-325 (10 patients) or CHEP (10 patients), Kwek et al. reported immediate hemostasis rates of 90% and 100% and definitive hemostasis rates of 67% and 90%, respectively [44]. However, only 8 (3 in the TC-325 arm and 5 in the CHEP arm) out of 20 patients (40%) had Forrest Ia or Ib PUD, while the remaining 12 had nonbleeding Forrest IIa or IIb PUD. As TC-325 may be active only in cases of active bleeding, these findings are difficult to assess. Holster et al. prospectively treated eight patients with PUD (four with Forrest Ia and four with Forrest Ib) with TC-325 as monotherapy (six patients), achieving immediate and definitive hemostasis rates of 83% and 67%, respectively [25]. The largest case series published to date is a prospective single-arm multicenter study of 202 patients with PUD-related NVUGIB, 156 patients with active bleeding (39 patients (19%) with Forrest Ia and 117 patients (58%) with Forrest Ib), and 46 patients with nonactive bleeding (25 patients (12%) with Forrest IIa and 21 patients (10%) with Forrest IIb) treated with TC-325 as combination therapy (101 patients), rescue therapy (51 patients), and monotherapy (50 patients), showing an overall immediate hemostasis rate of 88% and a definitive hemostasis rate of 71% [41]. According to its application, TC-325 achieved immediate and definitive hemostasis rates of 89% and 74% as combination therapy, 86% and 64% as rescue therapy, and 88% and 72% as monotherapy. Taking into account only patients with active bleeding, the immediate hemostasis rate was 85% in the Forrest group (87% Forrest Ia and 85% Forrest Ib); however, only a small proportion of these patients (≤25%) received TC-325 as monotherapy, and the outcome was not specified.

Two prospective studies on TC-325 administered as monotherapy in PUD-related NVUGIB were very recently published. In a prospective, single-arm, multicenter study, Sung et al. evaluated the efficacy of TC-325 as monotherapy in 67 patients with actively bleeding PUD (11 patients (16%) with Forrest Ia and 55 patients (84%) with Forrest Ib) who had not already undergone another HEP [47]. Patients received up to three canisters of TC-325 (20 g per canister) and, if unresponsive, were treated with a CHEP according to the physician’s preference. Persistent or recurrent bleeding within the first 72 h (early rebleeding) was the primary endpoint; recurrent bleeding between 72 h and 30 days (late rebleeding), adverse events, and mortality within 30 days were additional outcome measures. In one patient, TC-325 was not administered due to the occlusion of two consecutive catheters, and CHEP was performed. TC-325 achieved initial hemostasis in 60/66 (91%) treated patients, and early and late rebleeding were endoscopically confirmed in 5 and 3 patients (7.6% and 4.5%), respectively, with an overall recurrent bleeding rate of 12.1% and a definitive hemostasis rate of 79%. Two patients with Forrest Ia in whom TC-325 achieved initial hemostasis died (mortality rate: 3%). Multiple logistic regression revealed that Forrest classification was the only variable associated with recurrent bleeding, where the highest risk was associated with Forrest Ia. In a noninferiority RCT by Lau et al. including 224 patients with NVUGIB due to malignant and nonmalignant conditions, 68 patients with PUD (all Forrest Ia or Ib) were randomized to TC-325 versus a combination of CHEPs (contact thermocoagulation or hemoclipping with or without prior injection of diluted epinephrine), with immediate hemostasis rates of 95.6% and 83.8% and definitive (30 days) hemostasis rates of 83.8% and 73.5%, respectively [46].

The latter two studies provide robust data in support of the efficacy of TC-325 when used as monotherapy. HP may be considered as one of the first-choice techniques, in addition to being used in combination with CHEPs or as rescue therapy, in the management of NVUGIB due to PUD-related oozing bleeding. In contrast, the efficacy of HP as monotherapy in inducing definitive hemostasis in NVUGIB due to Forrest Ia PUD still seems to need definitive validation by further research, as already suggested [19], ideally in RCTs including large populations. Spurting active bleeding due to Forrest Ia PUD is much less common than oozing [21,47,62]. However, it may be massive and life-threatening, especially in patients with clinical conditions compromised by other comorbidities and hemodynamic instability, and is likely the main reason for the recurrence of NVUGIB [21,47]. In this context, in our opinion, CHEPs remain the gold standard approach to Forrest Ia PUD except when the lesion is not easy to reach, the operator is not familiar with CHEPs, or there is a risk of perforation in the case of contact-based thermal procedures. This risk seems to be increased in patients with recurrent bleeding when treated with a second consecutive thermal contact therapy, such as heater probe [19,63]. As alternative forms of hemostatic therapy are suggested in the event of rebleeding if thermal contact was used at the initial endoscopy [19], HP may be a possible choice. In the absence of conditions in which HP seems a suitable approach, it is reasonable to question why HP should be used to treat a bleeding lesion that can be controlled by a CHEP. Another reason to use HP only when necessary is the high cost.

### 4.2. Is HP a Possible Gold Standard in Malignancy-Related NVUGIB?

There is currently no gold-standard treatment for MR-NVUGIB, and the choice of CHEP depends on the characteristics of the tumor, such as location, size, consistency of surface, and pathological angiogenesis. Malignant lesions often have a friable surface with multiple bleeding points, negatively affecting the effectiveness of CHEPs, and mechanical contact-based HEP carries the risk of worsening the bleed or perforation. Nonlesion-related conditions influencing hemostasis include underlying coagulopathy, disease burden, and severity of hemorrhage [6,61]. In these cases, the success of CHEPs in MR-NVUGIB is variable, with immediate hemostasis achieved in about 30–40% of patients but with a short-term rebleeding rate of about 40–80% and 90-day mortality of about 95%, mainly due to the preterminal stage [36,64]. Based on the possibility of its application on large surfaces, argon plasma coagulation is frequently used in clinical practice but with temporary efficacy and high recurrent bleeding rates in MR-NVUGIB [65,66]. HP could be the ideal procedure of choice in the management of MR-NVUGIB, as it acts in the absence of mechanical contact and can also be sprayed over a large surface, allowing the simultaneous treatment of multiple bleeding points. To date, data regarding the use of HP in patients with MR-UGIB derive mainly from retrospective studies analyzing heterogeneous series (mixed lesions) and prospective studies lacking controls (Table 2). Hussein et al. [43] evaluated the efficacy of TC-325 in a prospective study including 105 patients who received HP either as monotherapy (67%) or in combination with or as rescue therapy after failure of CHEPs (25%). The decision to use TC-325 was at the discretion of the endoscopist. Overall, TC-325 achieved immediate hemostasis in 97% of patients, with a 30-day rebleeding rate of 15% and a definitive hemostasis rate of 82%. Immediate and definitive hemostasis rates were 100% and 85%, respectively, when TC-325 was used as monotherapy and 88% and 70% when used as combination therapy. Regarding possible factors influencing rebleeding due to malignancy, a univariable analysis conducted by Pittayanon et al. found that poor performance status, an Eastern Cooperative Oncology Group (ECOG) score ≥ 3, and INR > 1.3 were significantly associated with increased early recurrent bleeding, while definitive hemostatic treatment subsequent to TC-325 use was predictive of less delayed recurrent bleeding [36]. However, in a multivariable analysis, no significant prognostic factor for delayed recurrent bleeding was identified [36]. In contrast, good performance status (ECOG score 0–2), cancer stage 1–3, and achievement of definitive hemostatic treatment (surgery, chemotherapy, radiotherapy, and embolization) were significantly associated with greater survival. In a retrospective cohort of 41 patients with MR-UGIB, classified as Forrest Ib in 93% of cases, Shin et al. evaluated the efficacy of UI-EWD (Nexpowder) as monotherapy (23 patients, 56%) or rescue therapy (18 patients, 44%) [57]. In this study, UI-EWD achieved overall immediate and definitive hemostasis rates of 97.5% and 67.5%, respectively; as monotherapy only, the immediate and definitive success rates were 100% and 73.9%, respectively. Our study of HP in MR-GIB involved 23 patients, 14 with primary and 9 with metastatic cancer, in a series including other conditions, all treated with HP or CHEPs [17]. The source of bleeding was the stomach in 15 patients, the duodenum in 6 patients, and the colon/rectum in 2 patients; in all cases, the bleeding was oozing from multiple sites. Considering only patients with UGIB, HP was used as monotherapy in 9 patients and as rescue therapy in 7 patients (TC-325 in 14 patients and Endoclot in 2 patients), with immediate hemostasis rates of 100% and 85.7% and definitive hemostasis rates of 50% and 100%, respectively, significantly higher than those in patients treated with CHEPs (immediate hemostasis: 41.7%; definitive hemostasis: 33.3%). Of the 21 patients with MR-UGIB, 16 had an advanced unresectable tumor, 11 were discharged for palliative treatment, and 5 died from MR complications other than GIB, while the 5 remaining patients underwent elective (4 patients) or emergency surgery (1 patient) due to failure of hemostatic procedures.

To date, only two RCTs have been published on the use of TC-325 in MR-NVUGIB. In a pilot RCT, Chen et al. compared the efficacy of TC-325 versus CHEPs in upper (17 patients) and lower (3 patients) GIB, with 10 patients allocated to each group; crossover to the other treatment was permitted in patients not responding to the assigned therapy [40]. TC-325 achieved immediate hemostasis in 9/10 patients (90%) when used as monotherapy and in 4/5 patients (80%) unresponsive to CHEPs (rescue therapy); the definitive hemostasis rate at 180 days was 70% and 40% in the TC-325 and CHEP group, respectively. In a very recent RCT, Martins et al. compared the efficacy of TC-325 in monotherapy versus CHEPs in 68 patients with NVUGIB due to primary or metastatic malignancy who had not already undergone any HEPs [48]. In the control arm, a CHEP was not mandatory but could be applied if the attending endoscopist judged that hemostatic treatment might benefit the patient. TC-325 was always used in the presence of active bleeding; if bleeding was nonactive but stigmata were present, endoscopic washing of the tumor surface with a water jet was performed to remove the clot and reactivate bleeding or to induce brisk bleeding so that TC-325 could adhere to the tumor surface and promote coagulation. Immediate hemostasis was achieved in 100% of patients treated in both groups, with not-statistically-different 30-day rebleeding rates (32.1% in the TC-325 group and 19.4% in the control group) and 30-day mortality rates (28.6% in the TC-325 group and 19.4% in the control group). The difference between long-term outcomes, specifically late rebleeding rates, in the two groups was not statistically significant, likely because this study was underpowered due to incomplete (62%) enrollment of the planned population. Nevertheless, this is the first RCT with a very large series of patients evaluating the efficacy of HP in MR-NVUGIB, and the findings support the hypothesis that HP is more effective than CHEPs.

Karna et al. very recently conducted a systematic review and meta-analysis of the literature aimed at assessing the efficacy of topical hemostatic agents (TC-325, Endoclot, ABS, and UI-EWD) in the management of MR-NVUGIB [67]. The authors excluded studies reporting variceal bleeding or other nontumoral bleeding, studies performed in pediatric populations, those reporting data in <10 patients, case reports, case series, letters to the editor, review articles, and editorials. In the case of overlapping cohorts, the study providing the most recent results and/or the study with the largest sample size was included. Due to these strict selection criteria, from an initial pool of 355 investigations, the final analysis included 16 studies, 2 RCTs [40,68], and 14 observational studies [20,21,30,33,36,37,38,42,43,51,54,57,69,70]. In this meta-analysis, immediate hemostasis was achieved in 94.1% of patients, early rebleeding was observed in 13.9%, and delayed rebleeding in 11.4%, with an aggregate rebleeding of 24.2%. A subgroup analysis revealed similar immediate hemostasis rates in the TC-325 and non-TC-325 cohorts (93.9% versus 96.7%, respectively). All-cause mortality was 33.1%, while GIB-related mortality was 5.9%. Despite the large number of patients included in the investigation, this meta-analysis has the limitation of including both retrospective and prospective studies, some of which combine bleeding and nonbleeding (adherent clot and visible vessel) lesions, and only two RCTs.

The paucity of studies comparing HP with CHEPs does not provide sufficient data to draw a definitive conclusion as to whether HP may be considered a gold-standard procedure in MR-NVUGIB. Multicenter comparative studies on larger populations are necessary to confirm this potential role and to determine in which kind of malignant lesions HP can be most effective. However, before further controlled findings are available, current data allow us to make some observations regarding the use of HP as monotherapy in MR-NVUGIB. First, as previously mentioned, HP can be used to simultaneously treat multiple bleeding sites in large lesions often difficult to reach or to treat with no risk of perforation as with the traumatic action of contact-based CHEPs. HP was highly effective in inducing immediate hemostasis in the majority of cases of NVUGIB in which it was used as monotherapy or rescue therapy. Second, it is possible that different types of HP, namely TC-325 and non-TC-325, may have similar efficacy in the treatment of MR-NVUGIB [45,67]. This hypothesis needs confirmation in comparative studies, particularly with newer types of HP that do not require active bleeding to perform their action. Third, HP is a temporary and palliative intervention in GIB related to advanced diseases, the so-called preterminal stage, and cannot affect the outcome. In malignancies, progressive tissue destruction and necrosis contribute to rebleeding GIB even following treatment with TC-325 [35]. However, HP may play a bridging role toward a more definitive management of diseases still amenable to treatment. Albeit for a short period, the achievement of hemostasis by HP, when used as monotherapy and to a much greater extent as rescue therapy, provides valuable time to stabilize the patient’s cardiovascular functions and/or recover coagulation parameters when impaired, make blood transfusions, and carry out elective surgery, radiological procedures, or other definitive interventions [35]. HP should therefore be considered a first-choice hemostatic technique, as it is highly effective in MR-NVUGIB.

## 5. Conclusions

HP is highly effective in patients with NVUGIB when used either in combination with or as rescue therapy in the event of failure of CHEPs. Recent robust evidence supports the efficacy of TC-325 as monotherapy in NVUGIB due to spurting or oozing (Forrest Ia and Ib) PUD, but it cannot be regarded as an alternative to CHEPs as first-line treatment. Additional randomized comparative studies are necessary to define the role of HP as monotherapy and to establish whether it may be considered a gold standard in MR-NVUGIB. The potential role of new formulations, such as UI-EWD, in preventing bleeding during postendoscopic procedures and rebleeding following an early HEP needs to be confirmed by further investigations.

## Figures and Tables

**Figure 1 medicina-59-00143-f001:**
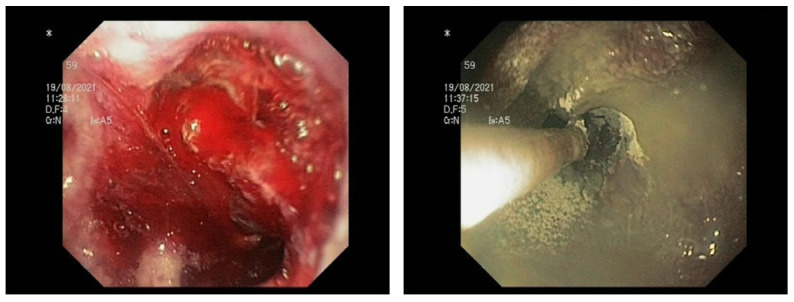
Duodenal cancer.

**Figure 2 medicina-59-00143-f002:**
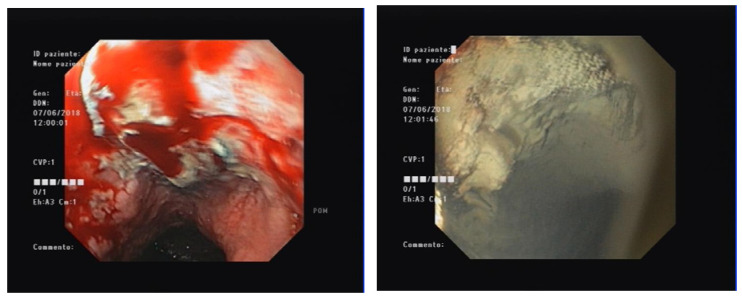
Esophageal cancer.

**Figure 3 medicina-59-00143-f003:**
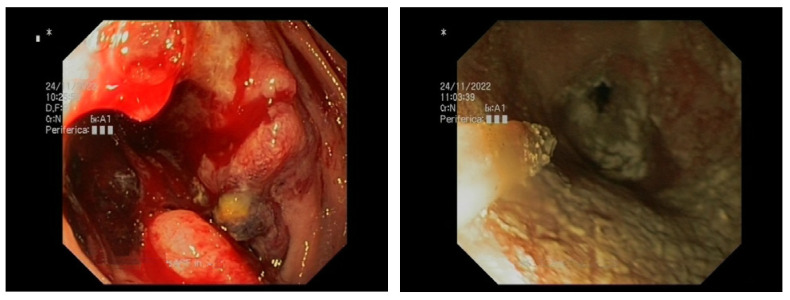
Pancreatic cancer invading duodenum.

**Table 1 medicina-59-00143-t001:** Hemostatic powders available on the market.

Trade Name	Market Area	Composition	Action
Hemospray	Canada, Europe, USA	inert mineral	absorption of water, promotion of clotting, coagulation cascade activation, mechanical tamponade
Endoclot	Turkey, Europe, Malaysia Australia	starch-derived polysaccharides	absorption of water promotion of clotting, coagulation cascade activation
Ankaferd Blood Stopper	Turkey	herbal ingredients	protein network promoting erythrocyte aggregation interaction with blood protein
Nexpowder	South Korea, Europe, USA	aldehyde dextran and succinic acid modified ε-poly (l-lysine)	adhesive hydrogel with multiple crosslinks within the hydrogel and between the hydrogel and tissue
CGBio	South Korea	hydroxyethylcellulose, EGF	adhesive seal in which the EGF promotes wound healing

EGF: epidermal growth factor; TGF: transforming growth factor.

**Table 2 medicina-59-00143-t002:** Studies on hemostatic powders and outcomes in patients with upper gastrointestinal bleeding.

Author, Year (Reference)	Country	Design	Hemostatic Powder	Indication	Forrest Ia/Ib (%)	Application	Hemostasis
Sung, 2011 [24]	Hong Kong	PC, N = 20	TC-325	PUD	5/95	Mono	I: 95% D: 95% (30 days)
Holster, 2013 [25]	The Netherlands	PC, N = 16	TC-325	MR-GIB, PUD, Other	31/25	Mono, Rescue	I: 81% D: 49.7% (7 days)
Leblanc, 2013 [26]	France	CS, N = 17	TC-325	MR-GIB, Other	NA	Mono, Rescue	I: 100% D: 88% (7 days)
Smith, 2014 [27]	France, Denmark, Germany Italy, Spain, Sweden, UK, The Netherlands	RC, N = 63	TC-325	MR-GIB, PUD, Other	17/25	Mono, Combo	I: 85% D: 70% (7 days)
Sulz, 2014 [28]	Switzerland	CS, N = 16	TC-325	PUD, Other	0/25	Mono, Rescue	I: 94% D: 81% (7 days)
Yau, 2014 [29]	Canada	RC, N = 19	TC-325	UGIB	21/57	Mono, Rescue	I: 93% D: 61% (7 days)
Chen, 2015 [30]	Canada	RC, N = 66	TC-325	UGIB, LGIB	7/23	Mono, Rescue	I: 99% D: 84% (30 days)
Giles, 2016 [31]	New Zealand	CS, N = 36	TC-325	PUD, Other	20/60	Mono, Rescue	I: 100% D: 89% (7 days)
Haddara, 2016 [20]	France	PC, N = 202	TC-325	MR-GIB, PUD, Other	7/21	Mono, Rescue	I: 96.5% D: 63.2% (30 days)
Sinha, 2016 [32]	UK	RC, N = 20	TC-325	PUD	60/40	Rescue, Combo	I: 92–100% D: 75–83% (7 days)
Arena, 2017 [33]	Italy	RC, N = 15	TC-325	MR-UGIB	NA	Mono	I: 93% D: 72% (6 days)
Cahyadi, 2017 [34]	Germany	RC, N = 52	TC-325	MR-GIB, PUD, Other	0/39	Mono, Rescue	I: 98% D: 51% (7 days)
Hagel, 2017 [35]	Germany	RC, N = 25	TC-325	MR-GIB, PUD, Other	ND	Mono, Rescue	I: 96% D: 60% (30 days)
Pittayanon, 2018 [36]	Canada	RC, N = 86	TC-325	MR-GIB	1/94	Mono, Rescue	I: 98% D: 72% (30 days)
Ramírez-Polo, 2019 [37]	Mexico	RC, N = 81	TC-325	MR-GIB, PUD, Other	ND	Mono, Combo	I: 99% D: 79% (5 days)
Rodriguez De Santiago, 2019 [21]	Spain	RC, N = 261	TC-325	MR-GIB, PUD, Other	25/64	Mono, Rescue	I: 93% D: 73% (30 days)
Meng, 2019 [38]	Canada	RC, N = 25	TC-325	MR-GIB	8/76	Mono, Rescue	I: 88%, D: 50% (14 days)
Alzoubaidi, 2020 [22]	France, Germany UK	PC, N = 314	TC-325	MR-GIB, PUD, Other	17/60	Mono, Combo, Rescue	I: 89% D:79% (3 days)
Baracat, 2020 [39]	Brazil	RCT, N = 19 N = 20	TC-325 CHEP	MR-GIB, PUD, Other	16/84 5/95	Mono, Combo	I: 100% D: 74% (7 days) I: 90% D: 75% (7 days)
Chahal, 2020 [23]	Canada	RC, N = 86	TC-325	MR-GIB, PUD, Other	14/53	Mono, Combo	I: 88% D: 55% (30 days)
Chen, 2020 [40]	Canada	RCT, N = 10 N = 10	TC-325 CHEP	UGIB, LGIB	NA	Mono, Rescue	I: 90% D: 70% (180 days) I: 40% D: NA
Hussein, 2021 [41]	France, Germany UK, USA	PC, N = 202	TC-325	PUD	19/58	Mono, Combo, Rescue	I: 88% D: 71% (30 days)
Becq, 2021 [42]	France	RC, N = 152	TC-325	UGIB, LGIB	ND	Mono, Rescue	I: 79% D: 39% (30 days)
Hussein, 2021 [43]	France, Germany Spain, UK, USA	PC, N = 105	TC-325	MR-GIB	NA	Mono, Combo, Rescue	I: 97% D: 82% (30 days)
Kwek, 2017 [44]	Singapore	RCT, N = 20	TC-325 CHEP	PUD	10/40 0/33	Mono	I: 90% D: 67% (4 weeks) I: 100% D: 90% (4 weeks)
Vitali, 2019 [45]	Germany	PC, N = 154	TC-325 EndoClot	MR-GIB, PUD, Other	11/66	Mono, Rescue	I: 81% D: 67% (30 days) I: 81% D: 56% (30 days)
Paoluzi, 2021 [17]	Italy	PC, N = 43	TC-325, Endoclot CHEP	MR-GIB, PUD	16/84 22/78	Mono, Rescue	I: 86–100% D: 45–86% (30 days) I: 42–78%; D: 33–69% (30 days)
Lau, 2022 [46]	Hong Kong, Thailand, Singapore	RCT, N = 224	TC-325 CHEP	MR-GIB, PUD, Other	8/92 11/89	Mono	I: 93%, D: 90% (30 days) I: 91% D: 81% (30 days)
Sung, 2022 [47]	Canada, Hong Kong, The Netherlands, UK	PC, N = 67	TC-325	PUD	16/84	Mono	I: 91% D: 78% (30 days)
Martins, 2022 [48]	Brazil	RCT, N = 59	TC-325 CHEP	MR-UGIB	NA	Mono	I: 100% D: 68% (30 days) I: 100% D: 80% (30 days)
Beg, 2015 [49]	UK	RC, N = 21	EndoClot	PUD, Other	24/76	Rescue	I: 100% D: 95% (30 days)
Prei, 2016 [50]	Germany	PC, N = 70	EndoClot	UGIB, LGIB	1/66	Mono, Rescue	I: 83% D: 72% (3 days)
Kim, 2018 [51]	South Korea	RC, N = 12	EndoClot	MR-GIB	NA	Mono, Rescue	I: 100% D: 84% (3–5 days)
Park, 2018 [52]	South Korea	CC, N = 30	EndoClot	UGIB	17/70	Mono, Combo	I: 97% D: 94% (30 days)
Hagel, 2020 [53]	Germany	RC, N = 43	EndoClot	UGIB	ND	Mono, Rescue	I: 100% D: 76% (1 day)
Kurt, 2010 [54]	Turkey	CS, N = 10	ABS	MR-GIB	NA	Mono	I: 100% D: 100% (7–48 days)
Karaman, 2012 [55]	Turkey	PC, N = 30	ABS	UGIB	ND	Mono, Combo	I: 87% D: 100% (7 days)
Gungor, 2012 [56]	Turkey	PC, N = 26	ABS	UGIB	15/85	Mono, Combo	I: 73% D: 53% (2 days)
Bang, 2018 [13]	South Korea	RCT, N = 35	CEGP-003 CHEP	UGIB	0/86 0/81	Mono	I: 89% D: 86% (3 days)
Park, 2019 [18]	South Korea	PC, N = 17	UI-EWD	UGIB	12/88	Rescue	I: 94% D: 75% (30 days)
Park, 2019 [11]	South Korea	RC, N = 56	UI-EWD	UGIB	0/64	Mono	I: 96% D: 92% (7 days)
Shin, 2021 [57]	South Korea	RC, N = 41	UI-EWD	MR-GIB	7l-93	Mono, Rescue	I: 97% D: 67% (28 days)

CS: case series; D: definitive; I: immediate; LGIB: lower gastrointestinal bleeding; MR-GIB: malignancy-related gastrointestinal bleeding; PC: prospective cohort; PUD: peptic ulcer disease; RC: retrospective cohort; RCT: randomized controlled trial; UGIB: upper gastrointestinal bleeding; NA: not applicable; ND: not defined.

## Data Availability

Not applicable.

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
