# Peer review of "Hemostatic Powders in Non-Variceal Upper Gastrointestinal Bleeding: The Open Questions"

_medicina, 2023, doi:10.3390/medicina59010143_

Round 1

Reviewer 1 Report

The review proposed by Paoluzi et al. is an interesting overview about current use and potential indication for hemostatic powders. Growing literature still lacks of recommendation and clear guidelines for everyday practice. This point is correctly discussed by the authors who report some "open questions".

I think the manuscript is interesting and gives an extensive review of current literature. I suggest a minor revision to improve it :

First, authors should declare if this is a narrative or systematic review. It is always recommended to report the search strategy showing literature choice and to add PRISMA flowchart in case of systematic review.

Then, I should add to "Pros" that there are no controindications to use these devices or specify if there are some reports about this (e.g. allergic reactions); "Cons" should include the problem that visibility is no more guaranteed after application.

Author Response

Reviewer 1

Many thanks for the appreciation and useful suggestions for improving our review article.

#1 - First, authors should declare if this is a narrative or systematic review. It is always recommended to report the search strategy showing literature choice and to add PRISMA flowchart in case of systematic review.

Re: Our review was narrative; we now specified this point in the Introduction (page 3 line 61). Therefore, we are not able to add a Prisma flowchart for showing the search strategy adopted.

#2 - Then, I should add to "Pros" that there are no controindications to use these devices or specify if there are some reports about this (e.g. allergic reactions); "Cons" should include the problem that visibility is no more guaranteed after application.

Re: We clarified the contraindications, adverse events, and Food and Drug Administration statement regarding TC-325 (Hemospray) and Endoclot (Page 5 lines 107-112).

Regarding Cons, we added a sentence highlighting the possibility of an obscured view following the use of HP (page 6 lines 123-124)

Reviewer 2 Report

The study, from a very rinomate endoscopic Italian center, is exaustive and well written. It is original and provides a wide range of information on a relative new thema. It is definitively worth the publication. 

Author Response

Reviewer 2

We are very grateful for your appreciation of our review article.

Reviewer 3 Report

hemostatic powders for non-variceal upper gastrointestinal bleeding. Although the manuscript is professionally written, there are some concerns that must be addressed. 

Major

  1. If some figures or images about hemostatic powder devices or endoscopic images during hemostasis are available, it helps readers to understand the manuscript.
  2. Although the authors showed the results of each important study about hemostatic powders, it is probably difficult for the authors to understand the list of numerical values in the manuscript. If this is a systematic value, we can understand the results of many studies visually by forest plot. Please summarize the studies in the open question section visually using the figures. 

Author Response

Reviewer 3

Many thanks for the suggestions useful for improving our review article.

# 1 - If some figures or images about hemostatic powder devices or endoscopic images during hemostasis are available, it helps readers to understand the manuscript.

Re: we added three sets of double images showing different lesions during the bleeding and after hemostasis achieved by the application of hemostatic powder

# 2 - Although the authors showed the results of each important study about hemostatic powders, it is probably difficult for the authors to understand the list of numerical values in the manuscript. If this is a systematic value, we can understand the results of many studies visually by forest plot. Please summarize the studies in the open question section visually using the figures.

Re: The review is narrative; we specified this point in the Introduction (page 3 line 61); thus we are unable to visualize the results of the studies in form of a Forest plot.

Round 2

Reviewer 3 Report

The authors revised the manuscripit appropriately.